# Establishment of an Enteric Inflammation Model in Broiler Chickens by Oral Administration with Dextran Sulfate Sodium

**DOI:** 10.3390/ani12243552

**Published:** 2022-12-15

**Authors:** Lixuan Liu, Wenjing Sui, Yajin Yang, Lily Liu, Qingqing Li, Aiwei Guo

**Affiliations:** 1Faculty of Life Sciences, Southwest Forestry University, No. 300, Bailong Road, Panlong District, Kunming 650224, China; 2Kunming Xianghao Technology Co., Ltd., Kunming 650204, China

**Keywords:** broiler, enteric inflammation, dextran sodium sulfate, growth performance, intestinal damage

## Abstract

**Simple Summary:**

Since the ban of antibiotic growth promoters (AGP) in animal feed, establishing an enteric inflammation model can help to accurately evaluate the effects of feed additives (e.g., prebiotics, probiotics, and organic acids) to maintain intestinal health and improve growth performance in broilers. The goal of this study was to assess the feasibility of using DSS (dextran sodium sulfate) to establish an enteritis broiler model. The results showed that DSS effectively induced enteric inflammation in broilers, as evidenced by poor growth performance, an increased inflammatory score, gross lesions, and severe histopathological damage in the jejunum.

**Abstract:**

This study aimed to evaluate the effectiveness of oral gavage of dextran sodium sulfate (DSS) to establish an enteric inflammation model in broilers. Forty 1-day-old male, yellow-feathered broilers were randomly divided into 2 groups with 5 replicates of 4 birds each for a 42-day trial. The experiment design used 2 groups: (1) the control group (CT), normal broilers fed a basal diet, and (2) the DSS group, DSS-treated broilers fed a basal diet. The DSS group received 1 mL of 2.5% DSS solution once a day by oral gavage from 21 to 29 days of age. The results showed that compared with those in CT, DSS treatment significantly increased histological scores for enteritis and mucosal damage at 29 and 42 days of age (*p* < 0.01) and the disease activity index (DAI) from 23 to 29 days of age (*p* < 0.01). DSS-treated broilers showed poor growth performance at 42 days of age, including decreased body weight and average daily gain and an increased feed conversion ratio (*p* < 0.01). DSS also caused gross lesions and histopathological damage in the jejunum of broilers, such as obvious hemorrhagic spots, loss of villus architecture, epithelial cell disruption, inflammatory cell infiltration, and decreased villus height. These results suggest that oral gavage of DSS is an effective method for inducing mild and non-necrotic enteric inflammation in broilers.

## 1. Introduction

Broiler enteric inflammation is a pathogen-induced intestinal condition that has resulted in a substantial economic loss for global poultry production because of poor growth performance and the costs of preventive and therapeutic measures [1]. In the post-antibiotic growth promoter (AGP) era, studies have explored nutritional strategies to prevent or alleviate enteric inflammation in broilers [2,3,4]. However, no broiler enteric inflammation model effectively mimics field situations; thus, research aimed at evaluating anti-inflammatory effects mediated by various feed additives has been limited [5]. Most methods that have been applied to induce broiler enteric inflammation are difficult to replicate in large-scale experiments owing to the complicated methodology and potential biosecurity risks, including lipopolysaccharide injection and challenges with *Clostridium perfringens* and *Eimeria maxima* [6]. Thus, establishing a secure, replicated, and simple model that can serve as an effective tool for studying enteric inflammation in broiler chickens is necessary.

DSS, a sulfated polysaccharide with anticoagulant activity, is widely used to induce colitis in mice and is a critical animal model for studying inflammatory bowel disease (IBD) and colitis-associated cancer in humans [7]. DSS-induced enteric inflammation may be related to various factors, for example, intestinal tissue histopathological damage, immune dysfunction, gut microbiota, and microbial metabolite dysbiosis. An in vitro study indicated that continuous DSS exposure led to a decrease in mucus thickness and an increased permeability of mouse colon tissue, allowing the bacteria to elicit inflammation through close contact with epithelial cells [8]. Goblet cell loss, epithelial cell disruption, crypt structure damage, and inflammatory cell infiltration are also common phenomena observed in DSS-induced colitis [9,10]. In addition, DSS downregulated the expression of tight junction proteins, impairing the intestinal mechanical barrier [11]. DSS-induced intestinal inflammation can be attributed to the activation of relevant signaling pathways, such as TLR4/TRIF/NF-κB [12], and this activation process may depend on the formation of nano-lipocomplexes through the linkage of DSS with medium-chain fatty acids [13]. Once the inflammatory signaling pathway is activated, many pro-inflammatory cytokines, including TNF-α, IL-1β, and IL-6, are detected in the serum or plasma [7,14]. Finally, dysbiosis of the gut microbiota and their metabolites is also involved in DSS-induced intestinal inflammation. For example, butyrate and butyrate-producing bacteria play a critical role in anti-inflammation, and their concentrations and relative abundance significantly decrease after DSS induction [7,9]. Moreover, bile acid metabolism dysbiosis, indicated by primary bile acid accumulation and secondary bile acid depletion, is closely related to DSS-induced intestinal inflammation progression [10,15]. The aforementioned studies suggest that intestinal barrier dysfunction is the primary cause of DSS-induced intestinal inflammation.

Accumulating studies have begun to use DSS to induce enteric inflammation in broilers due to its controllability [16]. Currently, two methods are used to treat DSS-induced enteric inflammation in broilers: drinking water [17] and oral gavage [6,18]. However, because of inconsistent water consumption, obtaining exact results in the experiment to establish enteric inflammation by DSS water administration was difficult. Therefore, oral gavage of DSS may be an appropriate option for inducing enteric inflammation in broilers. Several studies have reported successfully developing a broiler enteric inflammation model by oral gavage of DSS, as evidenced by increased serum FITC (Fluorescein Isothiocyanate Dextran) levels [19] and microscopical lesions scores in the duodenum, jejunum, and ileum [6]. However, some data, such as growth performance, intestinal development, clinical signs, and impact at the later stage of broiler development are unclear. In the present study, we explored the effect of oral gavage of DSS on growth performance, clinical signs, intestinal development, intestinal lesions, and histological change in broilers. We hypothesized that oral gavage of DSS could be used as an effective way to induce enteric inflammation in broilers.

## 2. Materials and Methods

### 2.1. Animal Management, Diets, and Experimental Design

Animal use and care were approved by the Academic Committee of the Southwest Forestry University (SWFU-20200715-1). Forty 1-day-old male Chinese yellow-feathered broilers (Lingnan, an improved meat-type breed) purchased from a local commercial hatchery were randomly divided into two groups with five replicates and four broilers per replicate. The experiment lasted for 42 days. The experiment was designed for two groups: a control group (CT) and a DSS-induced group (DSS). The CT group was fed a basal diet without DSS oral gavage, while the DSS group was fed a basal diet with DSS oral gavage from 21 to 29 days of age. The diet ingredients and nutritional levels are listed in Table 1. During the entire experiment, the broilers had free access to food and water. The lighting schedule consisted of 24 h of light per day. The temperature in the initial three days was controlled at 36 °C and 35 °C and subsequently decreased by 2 °C per week until the final temperature of approximately 23 °C was maintained. The experimental scheme for DSS-induced enteric inflammation in broilers is shown in Figure 1.

### 2.2. Modeling Broiler Enteric Inflamamtion by DSS

The DSS (MW: 50 kDa) used in this study was purchased from Shanghai Yuanye BioTechnology Co., Ltd. (Shanghai, China). The broilers in the DSS group were gavaged a DSS solution (1 mL, 2.5% weight/volume) at the same time daily from 21 to 29 days of age. The major reason for the selected treatment schedule was due to the fact that broiler enteric inflammation usually occurs between two to six weeks of age. The volume [19] and concentration [6] of the DSS solution used in this study partly refer to the previous two studies.

### 2.3. Sample Collection

At days 30 (1 day post-treatment) and 43 (2 weeks post-treatment), 10 broilers (2 broilers per replicate) per group were sacrificed by exsanguination.

The entire jejunum, ileum, and cecum of five birds were collected for the measurement of intestinal development; subsequently, the jejuna of five birds were opened lengthwise and washed with ice-cold physiologic saline to observe gross lesions. Approximately 2 cm of the middle segments of the jejunum from the remaining five birds were cut off carefully and fixed in a 4% polyformaldehyde solution for 24 h for subsequent histomorphology analysis.

The sample sizes of DAI scores, BW change, ADG, ADFI, FCR, and intestinal development parameters were five replicates, while the sample sizes of BW, jejunal histology scores, VH, CD, and VH:CD were 10 birds.

### 2.4. Criteria for DAI and Jejunal Histological Scores

The DAI was a composite score to assess the severity of enteric inflammation, which combined the percentage of weight loss, fecal quality, and fecal blood [20]. The DAI scores were calculated from 21 to 29 days of age, and the final scores were the sum of three index scores: body weight loss, fecal quality, and fecal blood. The jejunal histological scores were the sum of 4 index scores, including the severity and extent of inflammatory cell infiltration, epithelial cell disruption, and villus height to crypt depth (VH:CD) ratio. The scoring systems for DAI and jejunal histology are shown in Table 2 and Table 3, respectively.

### 2.5. Jejunal Gross Lesions

Referring to the results of previous studies [21], the severity of gross lesions on the jejunal mucosa was determined according to the number of hemorrhagic spots. Table 4 shows the criteria for jejunal gross lesions.

### 2.6. Broilers Growth Performance Parameters

Body weight (BW) was recorded daily from 21 to 29 days of age and then at day 1 and week 2 post-treatment, respectively. From 21 to 29 days of age, BW was continuously recorded to monitor body weight changes caused by DSS. Total BW and feed intake were recorded to calculate average daily gain (ADG), feed conversion ratio (FCR), and average daily feed intake (ADFI) according to the method proposed in a previous study [22].

### 2.7. Hematoxylin-Eosin (HE) Staining

Fresh intestinal jejunum tissues were fixed with 4% polyformaldehyde solution for 24 h, and subsequent HE staining was conducted by Wuhan Servicebio Biotechnology Co., Ltd. (Wuhan, China). Microscopic examination and image acquisition were conducted using an orthotopic light microscope (Nikon Eclipse E100, Nikon, Japan) and an imaging system (Nikon DS-U3, Nikon, Japan). The photomicrographs of thee sections per broiler were taken at 40× and 100×, respectively. Ten sections (magnification of 40×) with clear intestinal structures per group were selected for the measurement of VH, CD, and VH:CD using ImageJ software (1.82u). The jejunal histological scores were obtained twice for each chicken on the basis of observations of sections.

### 2.8. Intestinal Development

The length of each intestinal segment was measured with a flexible tape to prevent unintentional stretching, and then the digesta of each intestinal segment were squeezed for subsequent intestinal weight measurement.

### 2.9. Statistical Analysis

The data were analyzed by an independent sample *t*-test using SPSS Statistics software (version 26.0) to compare the average differences between the two groups. Statistical significance was set at *p* < 0.05 (* *p* < 0.05; ** *p* < 0.01).

## 3. Results

### 3.1. Effect of Oral Gavage of DSS on Signs of Enteric Inflammation

From 23 to 29 days of age, approximately 13 broilers in the DSS group showed severe clinical signs, as shown by watery feces with gas and bloody feces. During the period of DSS induction, broilers in the DSS group showed higher DAI scores (*p* < 0.01) than those in the CT group (Figure 2A), and no significant differences were observed in BW loss between the two groups (*p* > 0.05) (Figure 2B). In addition, jejunal histological scores were significantly increased in the DSS group at 29 and 42 days of age (Figure 2C) (*p* < 0.01).

### 3.2. Effect of Oral Gavage of DSS on the Growth Performance of Broilers

At 42 days of age, oral gavage of DSS led to a decrease in BW and ADG (*p* < 0.01) but an increase in FCR (Figure 3A–C), meaning that birds were less efficient at feed conversion and in a diseased state (*p* < 0.01). There were no significant differences (*p* > 0.05) in BW, ADG, FCR, or ADFI at 29 days of age between the two groups (Figure 3A–D).

### 3.3. Effect of Oral Gavage of DSS on Jejunal Lesion and Morphology

To investigate the short- and long-term effects of oral gavage of DSS on jejunal lesions and morphology, we conducted gross lesion assessment and HE staining at two time points. We found that jejunum gross lesions in the DSS group were severe, as evidenced by multiple hemorrhagic spots (Figure 4A,D). The jejunum in the DSS group showed severe pathological changes, with epithelial cell disruption (red arrow) and inflammatory cell infiltration (green arrow) (Figure 4B,E). Furthermore, oral gavage of DSS led to a decline in jejunal VH and VH:CD (*p* < 0.01) but did not affect jejunal CD (*p* > 0.05) (Figure 4C,F).

### 3.4. Effect of Oral Gavage of DSS on Intestinal Development

To determine the short- and long-term effects of oral gavage of DSS on intestinal development, we measured the length, weight, and length-to-weight ratio in different intestinal segments at two time points. At 29 days of age, the length of the jejunum and cecum in the CT group was higher than that in the DSS group (*p* < 0.05) (Figure 5A). In addition, there were no differences observed in intestinal development (*p* > 0.05) (Figure 5B–F).

## 4. Discussion

Since the use of AGP has been banned, establishing an enteric inflammation model can help in determining new dietary formulations or feed additives to improve growth performance and gut health in broilers with enteric inflammation. In this study, to explore the effectiveness of DSS in inducing enteric inflammation in broilers, we administered DSS solution by oral gavage to yellow-feathered broilers. The results showed that enteric inflammation in broilers was successfully induced, as evidenced by the inflammatory signs, poor growth performance, jejunal histopathological damage, and severe jejunal lesions.

DSS is widely used to induce rodent colitis, which is a critical model for studying the pathogenesis of IBD in humans [23]. Mice receiving DSS via drinking water showed typical colitis signs, such as diarrhoea, hematochezia, and higher BW loss, which were displayed by increased DAI scores [9]. In this study, we observed increased DAI scores in the DSS group during the DSS administration period, which was consistent with the literature on DSS-induced colitis in mice [20] and pigs [24], confirming the successful establishment of DSS-induced enteric inflammation in broilers. However, during the period of DSS induction, there was no significant loss in the BW of broilers according to daily BW records, which is inconsistent with the results of studies on colitis in mice [7,25]. One potential reason for this discrepancy may be that the DSS dose used in our study induced low-grade enteric inflammation rather than acute enteric inflammation. By contrast, mice with acute experimental colitis usually exhibit BW loss during DSS administration [9,26]. The histological score is a critical tool for estimating the severity of broiler enteric inflammation in a more dynamic manner than by the linear measurements of villi and crypts [27]. Our results revealed that the histological scores in the jejunum of the DSS group were higher at 29 days of age, and after 14 days of recovery, the jejunal histological scores remained higher than those in the CT group. Similar results were reported by Dal Pont et al. [6], indicating that DSS-induced intestinal tissue histopathological damage is irreversible and persistent. The explanation for this result is that chickens seemed to be more sensitive to oral DSS than rodents [28]. Additionally, during the DSS-induction period, signs of bloody stools and diarrhoea were observed, which were similar to those of DSS-induced murine colitis. Zou et al. [29] and Menconi et al. [28] have reported the same clinical phenomena in DSS-induced enteric inflammation in broilers, and they indicated that the severity of the clinical phenomena was positively related to the DSS dose used in the experiment. Therefore, the DSS dose used in this study was appropriate. Bloody stools may be related to the anticoagulant properties of DSS [30], and cellular hemostasis dysfunction [31], and diarrhoea may be associated with the epithelial sodium channel (ENaC) and aquaporins (AQPs), which are involved in the leak-flux mechanism responsible for regulating water reabsorption in the colon [32,33]. Therefore, impaired water reabsorption may be the primary cause of diarrhoea. This view is supported by research conducted by Huang et al. [10], who reported that the mRNA expression of *ENaC* and *AQP*s was downregulated in the DSS-induced mouse colon.

Poor growth performance in broilers caused by enteric inflammation has been widely reported [34,35], which is one of the most important causes of economic loss [36]. In this study, relative to that in the CT group, DSS resulted in a decrease in the growth performance of broilers at 42 days of age, as shown by lower BW and ADF and higher FCR. These results are consistent with those in the literature. Zou et al. [29] explored the effects of DSS doses (0.75, 1.25, 1.5, and 2.5%) on enteric inflammation in broilers; the results showed that except for the 0.75% dose, the remaining 3 doses reduced BW compared with that in the CT group (406.3 g, 396.7 g, and 296.7 g vs. 455.8 g), suggesting that the negative effects of DSS on broiler growth performance were dose-dependent. Notably, another study indicated that broiler BW in the DSS group (526.87 g) was lower than that in the CT group (632.39 g), and this depressed growth performance was not alleviated by supplementation with sodium butyrate [37]. In a recent study [38], a more depressed growth performance was observed in broilers in the starter phase under DSS challenge, including lower BW (278.7 g vs. 368.6 g) and ADG (16.8 g vs. 23.2 g) and higher FCR (1.5 vs. 1.1), than in the CT group. Another study reported that when broilers were administered 2.5% DSS for 7 consecutive days from 15 to 21 days of age, the BW (573.83 g vs. 692.83 g) and ADG (20.43 g vs. 37.79 g) at 21 days of age were significantly lower than those belonging to the CT group [39]. Poor growth performance in broilers challenged with DSS may be attributed to several factors, including increased intestinal permeability caused by gut barrier dysfunction [40], lower nutrient absorption ability caused by reduced small intestinal apical hydrolase activities [41], liver inflammation and lipid metabolism disfunction [42], and immune stress that shifts dietary nutrients from growth processes to support various inflammatory immune responses and mediator syntheses [22]. Gut microbiota dysbiosis also inhibits growth performance. In particular, upregulated *Desulfovibrio* was the main sulfate-reducing bacteria, utilizing DSS with 17% sulfate groups to generate H_2_S, which could cause poor growth performance by impairing intestinal epithelial cells [43].

Intestinal morphology and lesions are important indicators of enteric inflammation. In this study, DSS led to severe jejunal histopathological damage and lesions at 29 to 42 days of age, as shown by epithelial cell disruption, inflammatory cell infiltration, a decrease in VH and VH:CD, and multiple hemorrhagic spots. Zou et al. [29] demonstrated that DSS resulted in evident histological changes, including goblet cell depletion, a thinned mucus layer, and a significantly shortened VH. Severe enterocyte epithelial cell proliferation and inflammatory cell infiltration were observed in the small intestine of broilers challenged with DSS [6], and intense proliferation of epithelial cells further aggravates the intestinal inflammatory response because these cells are immature [44]. By contrast, Chen et al. [38] did not find significant histological changes in the duodenum, jejunum, or ileum between the broilers in the CT and DSS groups. We speculated that the varying responses of broilers to DSS seemed to depend on DSS concentrations and molecular weight, host age, sex, and gut microbiological environment [16]. In addition, we observed multiple hemorrhagic spots in the jejunal mucosa of the broilers. Similarly, multiple hemorrhagic spots in the jejunal mucosa and ileal mucosa were observed in broilers with subclinical necro-hemorrhagic enteritis challenge [21], suggesting that hemorrhagic spots can be regarded as an intuitionistic marker of intestinal damage. The shortened colon length is a critical characteristic in DSS-induced colitis mice [45]. In the present study, after oral gavage of DSS for 9 consecutive days, the length of the jejunum and cecum was reduced at 29 days of age but then returned to normal levels at 42 days of age. A study demonstrated that sodium butyrate increased the length of the jejunum, ileum, and cecum [46], and the potential mechanism may be related to the beneficial effects of butyrate on the proliferation, differentiation, and maturation of epithelial cells by regulating gene expression and protein synthesis [47,48]. Because of the importance of butyrate in intestinal development, we speculated that DSS-mediated changes in the composition and function of gut microbiota, especially lower butyrate-producing bacteria abundance and decreased butyrate concentrations, were the leading causes affecting intestinal development. DSS-induced colitis mice had lower butyrate-producing bacteria abundance and butyrate concentrations than normal mice, as described previously [49,50]. Although we did not investigate the changes of gut microbiota and microbial metabolites caused by oral gavage of DSS in the yellow-feathered broilers, their importance in broilers enteric inflammation led us to hypothesize the potential cause of DSS-induced enteric inflammation [51].

## 5. Conclusions

In conclusion, we demonstrated that oral gavage of DSS effectively induced enteric inflammation in yellow-feathered broilers, as indicated by severe clinical signs, poor growth performance, damage to the jejunal structure, and gross lesions in the jejunal mucosa, and these negative effects persisted following DSS treatment discontinuation. Despite the fact that the present study did not explore the full range of inflammatory parameters or the gut microenvironment affected by DSS, it did prove that the DSS-induced enteric inflammation model was suitable to explore pathogenic mechanisms and test the efficacy of nutritional intervention in broilers with enteric inflammation.

## Figures and Tables

**Figure 1 animals-12-03552-f001:**
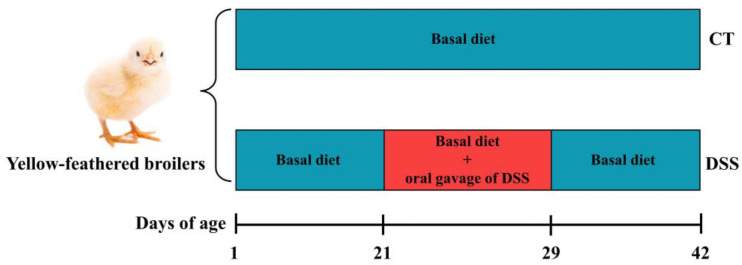
Experimental scheme of DSS-induced enteric inflammation in broilers.

**Figure 2 animals-12-03552-f002:**
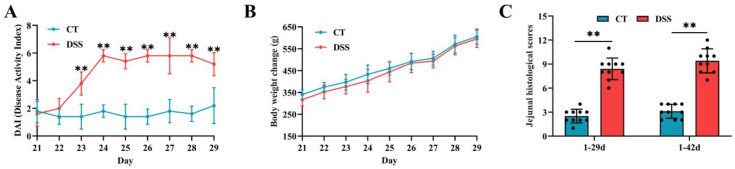
Effects of oral gavage of DSS on the signs of enteric inflammation in broilers: (**A**) DAI scores (n = 5 per group); (**B**) body weight changes (n = 5 per group); and (**C**) jejunal histology scores (n = 10 per group) at two different time points (1–29d and 1–42d). Values are means ± SD. ** *p* < 0.01. DAI: disease activity index.

**Figure 3 animals-12-03552-f003:**
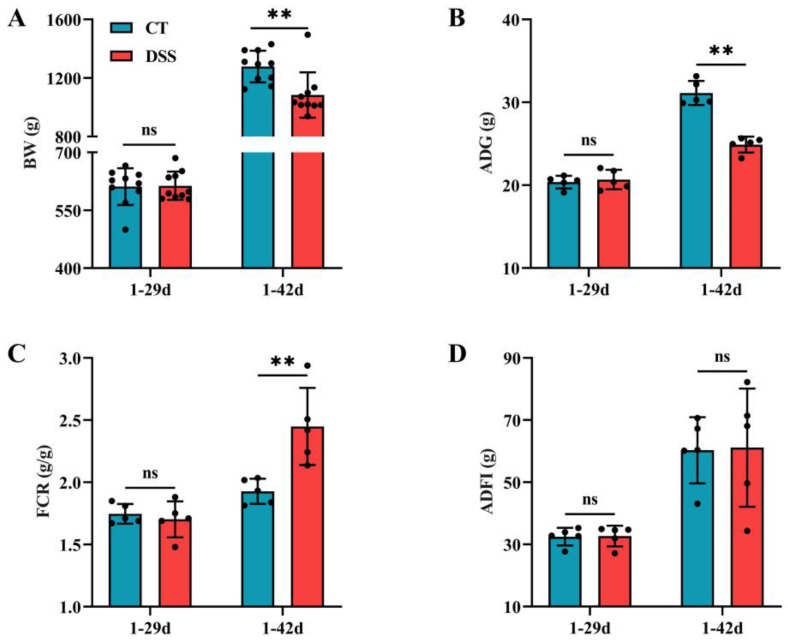
Effects of oral gavage of DSS on the growth performance of broilers: (**A**) BW at two different time points (1–29d and 1–42d) (n = 10 per group); (**B**) ADG at two different time points (1–29d and 1–42d) (n = 5 per group); (**C**) FCR at two different time points (1–29d and 1–42d) (n = 5 per group); and (**D**) ADFI at two different time points (1–29d and 1–42d) (n = 5 per group). Values are means ± SD. ** *p* < 0.01; ns: not significant (*p* > 0.05). BW: body weight; ADG: average daily gain; FCR: feed conversion ratio; ADFI: average daily feed intake.

**Figure 4 animals-12-03552-f004:**
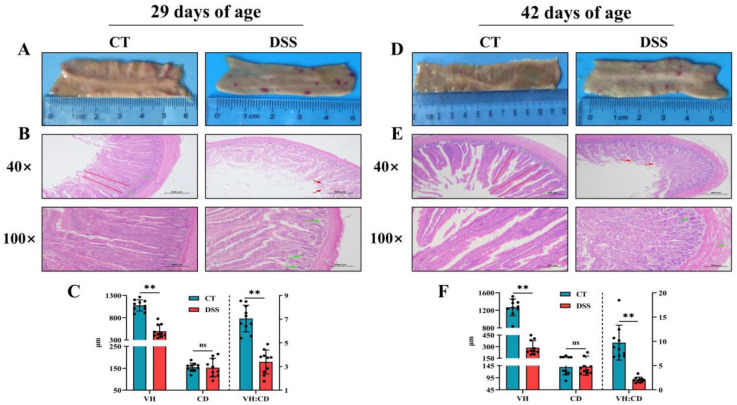
Effects of oral gavage of DSS on the jejunal lesion and morphology in broilers: (**A**) image of jejunum gross lesions at 29 days of age; (**B**) jejunal morphology by HE staining at 29 days of age; (**C**) VH, CD, and VH:CD of jejunum at 29 days of age (n = 10 per group); (**D**) image of jejunum gross lesions at 42 days of age; (**E**) jejunal morphology by HE staining at 42 days of age; and (**F**) VH, CD, and VH:CD of jejunum at 42 days of age (n = 10 per group). Values are means ± SD. ** *p* < 0.01; ns: not significant (*p* > 0.05). Red line, villus height; Green line, crypt depth; Red arrow, tissue disruption; Green arrow, inflammatory cell infiltration. VH: villus height; CD: crypt depth; VH:CD: ratio of villus height to crypt depth. CT = control; DSS = treated.

**Figure 5 animals-12-03552-f005:**
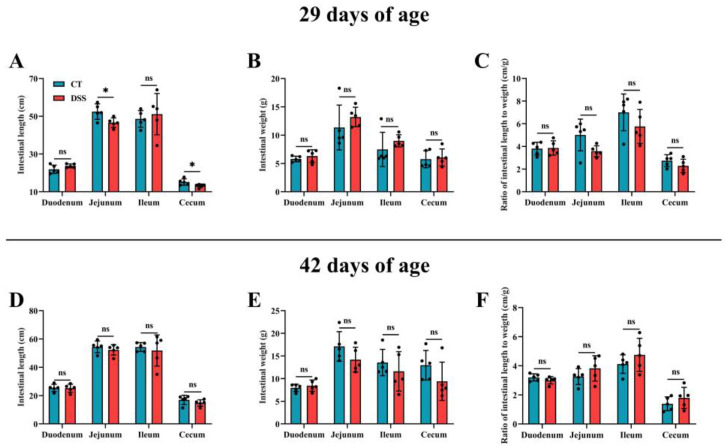
Effects of oral gavage of DSS on the intestinal development in broilers: (**A**) intestinal length at 29 days of age (n = 5 per group); (**B**) intestinal weight at 29 days of age (n = 5 per group); (**C**) ratio of intestinal length to weight at 29 days of age (n = 5 per group); (**D**) intestinal length at 42 days of age (n = 5 per group); and (**E**) intestinal weight at 42 days of age (n = 5 per group), (**F**) ratio of intestinal length to weight at 42 days of age (n = 5 per group). Values are means ± SD. * *p* < 0.05; ns: not significant (*p* > 0.05).

**Table 1 animals-12-03552-t001:** Ingredients and Nutritional Level of the Basal Diets (%).

Ingredients		Nutritional Levels ^2^	
Corn	54.19	Metabolism energy (MJ/kg)	12.12
Soybean meal	28.20	Crude protein	21.02
Fish meal	5.00	Lysine	1.30
Corn gluten meal	2.00	Methionine	0.56
Bentonite	3.00	Calcium	1.00
Soybean oil	3.00	Available phosphorus	0.48
L-Lysine	0.14		
DL-Methionine	0.19		
Limestone	1.50		
Dicalcium phosphate	1.27		
Sodium chloride	0.25		
Choline chloride	0.26		
Vitamin and mineral premix ^1^	1.00		

^1^ The vitamin and mineral premix provided per kilogram of diet: Cu, 8 mg; I, 0.70 mg; Fe, 100 mg; Mn, 120 mg; Se, 0.20 mg; Zn, 100 mg; vitamin A, 8000 IU; vitamin D3, 1000 IU; vitamin E, 20 IU; vitamin K3, 0.50 mg; vitamin B1, 2 mg; vitamin B2, 8 mg; vitamin B6, 3.50 mg; vitamin B12, 0.01 mg; vitamin H, 0.18 mg; nicotinamide, 35 mg; folic acid, 0.55 mg; pantothenic acid, 10 mg. ^2^ The nutritional levels were calculated values.

**Table 2 animals-12-03552-t002:** Scores of DAI in the Broiler Chicken DSS-Induced Enteritis Model.

Score	Severity	Body Weight Loss (%)	Fecal Quality	Fecal Blood
0	None	0	Normal	Normal
1	Mild	1–5	Soft feces	Blood on the fecal surface
2	Moderate	5–10	Irregular shaped feces	Blood on the fecal inner
3	Severe	>10	Watery feces with gas	Bloody feces

DAI: disease activity index.

**Table 3 animals-12-03552-t003:** Scores of Jejunal Histology in the Broiler Chicken DSS-Induced Enteritis Model.

Score	Inflammatory Cell Infiltration		Jejunal Structure Damage	
	Severity	Extent	Tissue Disruption	VH:CD Ratio
0	None	None	None	>6
1	Mild	Mucosa	Mild	4–6
2	Moderate	Submucosa	Moderate	2–4
3	Severe	Muscle layer	Severe	<2

VH: villus height; CD: crypt depth; VH:CD Ratio: ratio of villus height to crypt depth.

**Table 4 animals-12-03552-t004:** Criteria for Jejunal Gross Lesions in the Broiler Chicken DSS-Induced Enteritis Model.

Criterion	Number of Hemorrhagic Spots
None	0
Mild	1–2
Moderate	2–5
Severe	5–10

## Data Availability

The data presented in this study will be available upon request from the corresponding author.

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
