# Peer review of "Establishment of an Enteric Inflammation Model in Broiler Chickens by Oral Administration with Dextran Sulfate Sodium"

_animals, 2022, doi:10.3390/ani12243552_

Round 1
Reviewer 1 Report (Previous Reviewer 1)
This study to evaluate the effectiveness of oral gavage of dextran sodium sulfate to establish an enteric inflammation model in broilers. The study is ethically acceptable. However, it does not contain sufficient novel data to justify publication in Animals. In addition, the number of birds as well as the methodology used to confirm the intestinal inflammation was sufficient to justify publication in Animals. Therefore, my decision is “not accepted” for publication in Animals.
Author Response
Please see the attachment

Reviewer 2 Report (New Reviewer)
Work was well-written and clear overall. Figures were appropriate for the data and showed clear trends to readers. The researchers compare findings from this study to prior colitis modeling research in mice, but did not assess colon or even other small intestinal tissues in broilers to see if DSS treatment affects the whole GI tract or also induces colitis in broilers. I understand why the goal was to induce "enteritis" in broilers, but the authors should address why only jejunal tissue was sampled and how they are able to compare between colitis and enteritis models across species. Authors also describe prior broiler chicken studies using DSS and compare results of present study to results of those studies, but authors do not describe or demonstrate how this current study is different from and adds to what was learned from other broiler DSS studies. This information must be presented to show the relevance and novel nature of the present research.
The following are recommendations for revision to enhance clarity and understanding. Only those items starred (***) must be addressed by authors. Remaining items are suggestions and can be modified at the authors' discretion.
Comments by line # below:
16. “Pathohistological” is more commonly referred to as “histopathological”. Please change throughout the manuscript.
22. Was dosage daily/once a day during treatment period?
23. after "histologic scores" ADD for enteritis and mucosal damage, etc. (otherwise it is not apparent to readers what types of lesions were being scored)
24. Define term “disease activity index”
26. Describe gross lesions first, then histo because this is the order commonly assessed at necropsy and in research
35. Unsure what is meant by “comprehensive” intestinal condition. Please rephrase for meaning.
50. Is there any concern for using DSS in chickens as far as withdrawal times if treated birds were to intentionally or accidentally enter the food chain? I know the birds from this study did not enter the food chain, but it may be worth mentioning whether future DSS modeling studies should be aware of potential to cause enteritis/colitis in people if later entering the food chain.
52. Rephrase to “and gut microbiota and microbial metabolite dysbiosis” without commas. Currently gut microbiota by itself has no meaning. Needs to be part of the larger phrase.
***71. Not sure hypothesis is directly testable (or tested) as written. Hypothesis: DSS-induced enteric inflammation may be similar to broiler enteric inflammation caused by a specific pathogenic infection. There was no comparison in this experimental design between the signs/lesions of DSS treatment and another specific pathogenic infection (viral enteritis/RSS, Necrotic Enteritis/clostridium +/-coccidia, etc.). The hypothesis statement does not fit with the experimental design which is essentially to determine whether DSS treatment does or does not induce enteric mucosal damage. Simplify or rephrase hypothesis since not testing DSS against other specific pathogens. Of note, specific pathogens of broilers at 3 weeks of age can be quite variable and can span crypt lesions, generalized enteritis, villous lesions, widespread tissue necrosis, dysbacteriosis without lesions, etc. While DSS treatment did induce lesions in the study, it did not directly match lesions of any one of the “specific pathogens” typical in broilers of this age.
82. Change “premature” to “juvenile”
91. I am unfamiliar with the composition of the “Academic Committee” of the Southwest Forestry University compared to U.S. Institutional Animal Care and Use Committees. What qualifications do members of this group have to assess animal health and welfare in research? Does the group include at least one veterinarian to assess appropriate animal care and use in research?
***92. Add or describe genetic line of broilers, since different genetic lines may react differently to DSS treatment.
***119. Were daily doses administered at the same time daily or did dosing time vary?
124. Define disease activity index for reader. Does this equate to Incidence of disease?
***128. Why were only jejunal samples collected and analyzed? Does DSS treatment also affect other intestinal segments? The colon (since colitis model in other animal species)? Need to explain why analysis was limited to a single region. Was the same segment/anatomic location in the jejunum sampled each time or were samples collected from different areas of jejunum across birds?
***Table 2 Title and all table/figure legends- Write out all abbreviations (ex. DAI, VH, CD, VH:CD, etc.) in full in all tables and figures so they can “stand alone” from the written text. Add “in broiler chicken DSS-induced enteritis model” to give full description of the nature of the study. Add this or similar study specific information to all table/figure titles.
Table 3: Epithelial cell (change to singular) disruption
***139. Need to explain why 30 and 43 days of age were selected as collection time points. Actually, rather than using days of age at this point, I would recommend stating as 1 day post-treatment and 2 weeks post-treatment which makes more sense to reader. Change to "days post-treatment" or similar throughout remainder of manuscript since this means more to the reader. It did not seem that you had collected any true pre-treatment negative controls but that tissue would be useful to compared to 1d post-treatment to see ensure any acute injury was from administration of DSS rather than background or natural inflammation in tissues. Would recommend adding a true negative control for future studies.
154. Rather than “We obtained 3 sections”, consider being more specific “We imaged” or “We captured photomicrographs of 3 sections each at 40x and 100x mag….”
***164. Were histologic specimens collected before or after “squeezing”? If squeezing occurred before sample collection, could this handling of specimens account for some of the changes in villus height or V/C ratio? How did you minimize artifactual damage to tissues from tissue handling?
***Section 2.7: Put the * and ** p values in parentheses or brackets. Right now these values appear as almost a sentence fragment. Were any birds/tissues excluded from analysis? In this section, it states n=5, but in Figure 2, jejunal histology scores are n=10. Need to describe how groups were combined for analysis or state when the larger sample size was used. Also, must describe use of averages for comparison. These factors are not described currently, but all figures use average scores/measurements.
173. Change “symptoms” to “signs.” Technically animals cannot have clinical symptoms because they cannot communicate what they are experiencing to clinicians.
Figure 2c. I am not sure the data point dots add to the meaning and interpretation of this figure. Please consider removing the dots to simplify the image.
187. Remind readers unfamiliar with poultry that an increase in FCR means that birds are less efficient at feed conversion…the effect expected in a diseased state.
201. Described “jejunal lesions in the DSS group were severe” but summarized elsewhere as mild. Please add information about the scores assigned at the time of analysis (range of low to high scores and average), then summarize as either minimal, mild, moderate, or marked/severe, but not more than one of the severity terms since each has a different meaning. Lesions cannot be mild and severe at the same time.
Figure 4. Add CT = control; DSS = treated
***218. Since you did not collect tissues daily from day 1 to day 29, you cannot state that the length was higher in control than in treated. Please rephrase to match your experimental design. Did you collect day 1 tissues and compare to later tissues? If so, please add this to methods.
237. Need to explain here or in intro why it is important in broilers to induce and study enteritis compared to colitis as in the mouse model. And tell why you did not also assess colonic tissues in the present study.
255. “Irreversible and persistent”- Is this similar to what is expected in natural disease healing? Or could this model actually induce worse damage and more long term damage than most natural disease in broilers?
***257. Need to describe the signs of bloody stools and diarrhea that were observed in your results section first. I know these are summarized with some of the scoring you performed, but you should also summarize for reader how many birds were clinically affected and showed clinical signs in each group, and what range of severity of clinical signs and lesions you observed. This information does not come across in “average” scores reported per group.
***259. MUST tell reader how your study is different from or adds to what is already known from other DSS studies in broilers. Otherwise this work appears to just repeat what others have already done.
307. “varying responses” Tell whether you feel DSS induction is truly reproducible since you describe variable results from prior DSS work.
315. Does “this study” refer to your study or to reference #47? Unclear to reader.
322. Seems odd to reader that you finish your results discussing butyrate contribution to these effects since you did not measure butyrate concentrations. Could you instead end the section summarizing your key findings from your study?
Author Response
Please see the attachment

Reviewer 3 Report (New Reviewer)
In this manuscript, Liu et al have reported an enteric inflammation model in broilers with oral administration of dextran sulfate sodium. Following treatment of birds with DSS, several parameters were investigated to determine the influence on the growth of birds, histopathological morphology and and intestinal health. The experiment in general is well designed and performed however the following revisions are required.
1. Gut inflammation induced by administration of DSS, both oral and in drinking water, has already been reported. Thus the concept used to establish the in vivo model in this manuscript is not novel. Refer to the publications that authors have cited as well - eg. Kuttappan et al., 2016, Dal Pont et al., 2021 etc. Consequently, justification provided in the current study (lines 76-85) is not strong. Line 80 should be paraphrased. The previous cited experiments are not "flawed".
Suggestion: Please provide a strong justification explaining the uniqueness of the study compared to published papers. Authors might highlight the breed that is used i.e. Yellow coloured broilers as t is the local breed raised for meat purpose in China. Secondly, impact of DSS treatment at the later stage of broilers was studied - it should be emphasized more.
2. English language revision is necessary.
Author Response
Please see the attachment

This manuscript is a resubmission of an earlier submission. The following is a list of the peer review reports and author responses from that submission.
Round 1
Reviewer 1 Report
This study investigated the Establishment of Broiler Enteric Inflammation Model by Oral Administration with Dextran Sulfate Sodium (DSS). The study is ethically acceptable and contains a few novel data to justify publication in Animals. In the present study, results showed that DSS could serve as an effective way to induce enteric inflammation in the broiler, as evinced by poor growth performance, increased inflammatory score, as well as severe morphologies changes and macroscopic lesion in the jejunum. However, I have some minor comments.
· Line 31…..
Replace the word “disease” with “condition”
· Line 72
It is mentioned that the duration of the experiment was 42 days. However, later in the text (95-96 lines), it is mentioned that jejunal histology scores were calculated on 43 days.
Reviewer 2 Report
The authors in this manuscript investigated the feasibility of using DSS (Dextran Sodium Sulfate) to establish an enteritis model in broilers. The results demonstrated that DSS could serve as an effective way to induce enteric inflammation in broiler chickens, as evidenced by poor growth performance, increased inflammatory score, severe morphologies changes and macroscopic lesion in the jejunum. Generally speaking, the manuscript is interesting, and the manuscript was well organized. The introduction, methods and materials, results and discussion were proper and related to the topic. There are many grammar errors throughout the manuscript, for example, the lack of complete sentences in Ln 49, Ln 112, Ln 221-222, Ln 238, Authors should carefully check the basic errors, including but not limited to other errors.
1). Ideally, the control group should have executed a similar treatment of oral gavage with vehicle solution (water or PBS or other solutions), in order to impact similar stress and possible damage by the technique to the control birds, even though other scientists in referred literature may not do so. The oral gavage sometimes may cause some damage and mortality, depending on the operator’s techniques.
2). Ln 112: Revise the sentence: the chickens were sacrificed by XXX methods on Day 42, and
3). Ln 175: Correct “given” should be “gave” or “administrate”.
4). Ln Ln 178, 229, 248: Oral gavage of DSS.
5). Ln 194: Add “ be” as : may be associated with…
6). Ln 228-232: The DSS model caused hemorrhagic spots/ inflammation, but not a necrotic lesion in necrotic enteritis (NE) case, even though NE has hemorrhagic spots as well. You may revise the speculation.
7). Ln 266: Revise: “…will be available upon requests…”.
8). Should revise the title similarly: Establishment of An Enteric Inflammation Model in Broiler Chickens by Oral Administration with Dextran Sulfate Sodium
